# Hadamax Encoding: Elevating Performance in Model-Free Atari

**Jacob E. Kooi**
jacobkooi92@gmail.com
Department of Computer Science
Vrije Universiteit Amsterdam

**Zhao Yang** *
z.yang3@vu.nl
Department of Computer Science
Vrije Universiteit Amsterdam

**Vincent François-Lavet**
vincent.francoislavet@vu.nl
Department of Computer Science
Vrije Universiteit Amsterdam

## Abstract

Neural network architectures have a large impact in machine learning. However, in the specific case of reinforcement learning, network architectures have remained notably simple, as changes often lead to small gains in performance. This work introduces a novel encoder architecture for pixel-based model-free reinforcement learning. The Hadamax (**Hada**mard **max**-pooling) encoder achieves state-of-the-art performance by max-pooling Hadamard products between GELU-activated parallel hidden layers. Based on the recent PQN algorithm, the Hadamax encoder achieves state-of-the-art model-free performance in the Atari-57 benchmark. Specifically, without applying any algorithmic hyperparameter modifications, Hadamax-PQN achieves an 80% performance gain over vanilla PQN and significantly surpasses Rainbow-DQN. For reproducibility, the full code is available on GitHub.

## 1 Introduction

Ever since reinforcement learning (RL) algorithms [53] surpassed human players on the Atari-57 benchmark [6, 37, 38], progress has been driven mainly by various algorithmic innovations [15, 50].

Compared with the field of supervised learning (SL), the deep learning components of RL have remained relatively simple, usually consisting of a few convolutional layers (for image-based tasks) followed by fully connected layers [38, 27]. So far, the most common encoder modification in image-based RL tasks has been the integration of a ResNet encoder [13], inspired by its wide use in supervised learning architec-

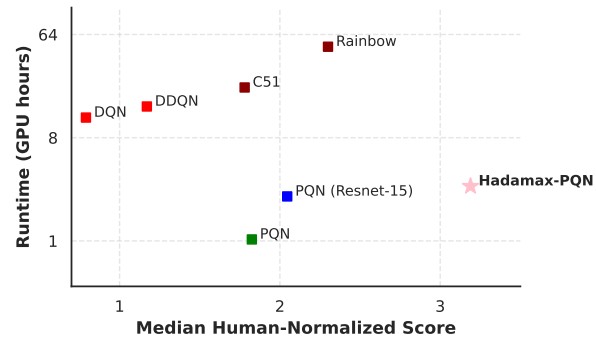

Figure 1: Performance versus GPU hours in the full Atari-57 domain at 200M environment frames. The application of our **Hada**mard **max**-pooling encoder on PQN yields significant performance improvements over a current state-of-the-art model-free method, Rainbow, while remaining more than an order of magnitude faster.

---

*Corresponding Author

39th Conference on Neural Information Processing Systems (NeurIPS 2025).

tures [25]. Several further approaches
have been explored to scale the deep learning architecture, but findings indicate that scaling pixel-based RL remains a significant challenge [41, 42], and finds greater success in either low-dimensional state-based continuous control [35, 24] or complex model-based architectures [47, 23].

In this work, we revisit the assumption that modifications to deep learning architectures can not lead to significant improvements in RL. We build on top of the recent Parallelized Q-Network (PQN), which reinvented DQN to function without the replay buffer and target network, while profoundly increasing performance [17]. This is done by combining recent advances in Hadamard representations [31] with max-pooling found in the ResNet encoder structures [25, 13]. Specifically, we augment the state-of-the-art PQN algorithm with a **Hada**mard-**max**pooling (Hadamax) encoder. The contributions can be summarized as follows:

- A novel deep learning architecture is proposed to improve usual pixel-based convolutional encoder architectures for model-free RL. This design shows an alternative direction of encoder synthesis in RL, as compared to the widely used deeper ResNet architectures.

- Without applying any algorithmic or hyperparameter modifications, Hadamax-PQN achieves an 80% performance gain in the full Atari-57 suite over the recent PQN baseline [17]. These changes allow Hadamax-PQN to significantly surpass Rainbow-DQN [27] while remaining more than an order of magnitude faster, setting a new state-of-the-art for model-free RL on Atari.

## 2 Related Work

Different neural network architectures are applied in RL to enhance the performance in online settings [37, 5, 13, 27, 17, 35] as well as offline limited data settings [7, 49, 10, 50, 39]. In this work, we focus on agents in the high-dimensional Atari-57 domain [6], a diverse and commonly-used challenging benchmark with discrete actions and pixel-based input.

**Network development in RL for Atari:** Deep Q-learning (DQN) [37, 38] achieves human-level performance on Atari games for the first time in the RL history by using three convolution layers (with ReLU) followed by fully-connected layers. Due to its simplicity and efficiency, this classic architecture is used for many later works, such as Double DQN (DDQN) [55], Dueling DQN [58], Noisy DQN [14], Categorical DQN (C51) [5] and Rainbow-DQN [27]. The recent Parallelized Q-Network (PQN) [17] algorithmically simplifies DQN and uses LayerNorm [4] to provably stabilize optimization. C51 [5] and R2D2 [29] enhance the output layer using categorical distributions and a recurrent network, respectively. In the context of model-based RL, Recurrent State-Space Models (RSSM) [20, 21, 23], image augmentation [34], forward prediction [49, 40], residual architectures [13, 46] and transformers [1] have also been explored to solve Atari. Impala [13] introduces a deeper ResNet-15 encoder structure with 6 residual blocks, allowing for high data efficiency under distributional training. BBF [50] further widens the Impala encoder, achieving state-of-the-art performance on the Atari-100k benchmark. SPR [49], using DQN's architecture with a self-prediction objective, also improves data efficiency. For model-based methods, residual architectures [60, 57], transformers [61] and diffusion models [3] are being increasingly leveraged to boost sample efficiency. Our work focuses on model-free agents in the Atari-57 benchmark, where relatively modest algorithmic architectures are used, and a large amount of environment interactions is allowed.

**Speedups in RL:** Since the development of JAX [9], parallel and vectorized training of reinforcement learning (RL) agents has become a promising area of research, offering significant performance and scalability improvements. Physics simulation engines and tools that are compatible with JAX have emerged to support this paradigm, including Brax [16], a physics simulation engine optimized for high-speed differentiable environments; Gymnax [33], a lightweight, JAX-based version of classic Gym environments; Jumanji [8], a suite of combinatorial and decision-making environments tailored for JAX; and EnvPool [59], a high-throughput environment execution engine with up to 20x speedup compared to Python. To complement these environments, a growing ecosystem of reinforcement learning libraries built entirely in JAX has been developed. PureJaxRL [36] implements standard RL algorithms entirely end-to-end in JAX, enabling parallel execution across thousands of environments. JaxMARL [45] focuses on multi-agent reinforcement learning, demonstrating strong acceleration of existing algorithms. Additionally, *cleanrl* [28], a library providing high-quality and reproducible RL

implementations, also includes several JAX-based implementations. Our work builds upon PQN [17], which leverages EnvPool and PureJaxRL, achieving greater efficiency compared to conventional PyTorch-based implementations. With the Hadamax encoder, we further architecturally improve PQN to the point that it significantly surpasses Rainbow-DQN, while remaining more than an order of magnitude faster.

## 3 Preliminaries

As a background, we briefly explain general value-based RL and the recent PQN algorithm, which is extended with our proposed encoder.

### 3.1 Reinforcement Learning and Value-based Methods

We consider a Markov Decision Process (MDP), defined by the tuple $< \mathcal{S}, \mathcal{A}, \mathcal{P}, \mathcal{R}, \gamma >$, with state space $\mathcal{S}$, action space $\mathcal{A}$, transition function $\mathcal{P}$, reward function $\mathcal{R}$ and discount factor $\gamma \in [0, 1)$. An agent in state $s_t \in \mathcal{S}$ at timestep $t$, taking action $a_t \in \mathcal{A}$ observes a reward $r_t \sim \mathcal{R}(s_t, a_t)$ and next state $s_{t+1} \sim \mathcal{P}(s_t, a_t)$. The goal is to learn an optimal policy $\pi^* : \mathcal{S} \to \mathcal{A}$ that can maximize the expected return $G(s_t) = \mathbb{E}\left[\sum_{k=0}^{\infty} \gamma^k r_{t+k} \mid s_t = s\right]$ over all possible trajectories. Unlike policy-based or actor-critic methods [48, 19] that optimize the policy, value-based methods [37] learn a state-action value function $Q(s, a)$. Once the optimal Q-function is learned, the optimal policy is implicitly defined by selecting greedy actions $\pi^*(s) = \mathrm{argmax}_a Q^*(s, a)$. Q-learning is the most widely used value-based algorithm. It learns $Q(s, a)$ through temporal difference (TD) learning. The update rule is:

$$Q(s_t, a_t) \leftarrow Q(s_t, a_t) + \alpha[r + \gamma \mathrm{max}_{a' \in \mathcal{A}} Q(s_{t+1}, a') - Q(s_t, a_t)], \tag{1}$$

where $\alpha$ is the learning rate. Over time, this iterative process allows the Q-function to converge to the optimal value function $Q^*(s, a)$, from which the optimal policy can be derived.

Deep Q-Network (DQN) [37] extends Q-learning by using a deep neural network to approximate the Q-function. The network is trained to minimize the difference between the predicted Q-values and the target values, typically using a loss function such as mean squared error:

$$\mathcal{L}(\theta) = \mathbb{E}_{s_t, a_t, r_t, s_{t+1} \sim D}[(r_t + \gamma \mathrm{max}_{a' \in \mathcal{A}} Q(s_{t+1}, a'; \theta^-) - Q(s_t, a_t; \theta))^2] \tag{2}$$

where $\theta$ and $\theta^-$ are the parameters of the Q-network and are the parameters of a target network that is periodically updated. $D$ is the experience replay buffer from which mini-batches are sampled.

### 3.2 Parallelized Q-Network (PQN)

PQN is a simplified deep online Q-learning algorithm. By parallelizing vectorized environments and normalizing neural networks (LayerNorm), PQN can stabilize the training even without a target network and replay buffer. Moreover, it is compatible with pure-GPU training, leading to efficient training on Atari tasks. More specifically, PQN makes the following modifications compared to the original DQN:

$\lambda$-**return**: Unlike the original DQN uses 1-step return, PQN leverages a more stable $\lambda$-return. The loss in Equation (2) thus becomes:

$$\mathcal{L}(\theta) = \mathbb{E}_{\mathrm{trajs}}[(r_t + \gamma(\lambda G_{t+1}^{\lambda} + (1 - \lambda)\mathrm{max}_{a' \in \mathcal{A}} Q(s_{t+1}, a'; \theta)) - Q(s_t, a_t; \theta))^2], \tag{3}$$

where $G^{\lambda}$ is the $\lambda$-return. When $\lambda = 0$ it will be similar to Q-learning, and if $\lambda = 1$ it is equivalent to the Monte Carlo update, which uses the full return until the episode ends.

**LayerNorm**: PQN adds LayerNorm for the output of convolution / MLP layers before the ReLU activation functions, which helps stabilize the training process.

**Removal of replay buffer and target network**: Since the whole training process happens on GPU, removing the replay buffer can largely reduce memory and thereby accelerate training. As a result of the training stability, the target network is also eliminated.

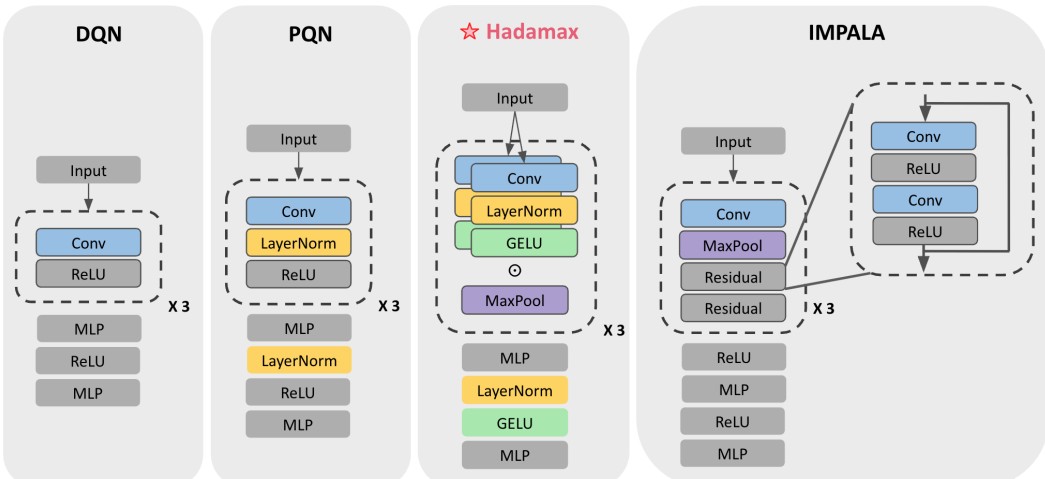

Figure 2: Encoder architectures of DQN, PQN , the proposed **Hada**mard **max**-pooling (Hadamax) encoder and the Impala ResNet-15 encoder (from left to right). In the Hadamax encoder, down-sampling is facilitated by max-pooling operators. Furthermore, we apply a Hadamard product between parallel representation layers. The implementation is straightforward and can be found in Appendix B. These changes allow for a substantial increase in algorithm performance, while keeping general encoder structure, convolutional depth and algorithmic hyperparameters unchanged.

# 4 Hadamax Encoder

The first human-level performance in the Atari-57 domain was achieved with the 'Nature' DQN encoder design [38]. The general effectiveness of this architecture, as well as the problems with scaling in deep RL, has led to this architecture's use even in the modern state-of-the-art algorithms such as PQN [17]. In this section, we provide the reasoning and implementation of the proposed **Hada**mard **max**-pooling augmentation of the original DQN encoder. For reproducibility purposes, we refer the reader to a detailed implementation of the proposed architecture in Appendix B.

## 4.1 Design Choice 1: Down-sampling by Max-pooling

As pixel-based observations are high-dimensional, the encoder must effectively compress the state representation to enable the downstream RL algorithm to converge within a reasonable number of updates. In the conventional DQN encoder, this compression is achieved by the convolutional operations (See Fig. 2), where the compression is determined by the convolutional kernel size and stride. In contrast, when examining the well-known and widely used Impala ResNet-15 encoder in RL [13], max-pooling is responsible for the bulk of feature compression. The resulting effect is that minimizing convolutional strides and adding max-pooling allows for the selection of a more dense representation of convolutional features, and subsequently emphasizes the strongest signals. Additionally, the use of max-pooling adds a slight translation invariance to the important features. We therefore hypothesize that the use of max-pooling in RL is, although widely implemented in supervised learning, relatively overlooked. In the Hadamax encoder, convolutional down-sampling is therefore replaced by max-pooling operators. Furthermore, in contrast to the average-pooling used by the original supervised learning ResNet architecture [25], the Hadamax encoder max-pools the final features before flattening to the linear layer. Since value functions in RL should be able to show strong correlations with the most important features, average-pooling before the linear layer will achieve the opposite, as it smoothens out feature importance.

The max-pooling design choices; max-pooling and downsampling instead of convolutional down-sampling, followed by max-pooling without down-sampling before flattening, are thus respectively influenced by the ResNet-15 (Impala) RL encoder and the original ResNet. However, in stark contrast to both residual encoders mentioned, the Hadamax encoder remains shallow (3 convolutional layers), and therefore no residual connections need to be applied.

## 4.2 Design Choice 2: Application of Hadamard Representations

Although multiplicative interactions have been commonly used in Deep Learning architectures [52, 56, 11], their application in RL remains limited. Recent work however has shown that the effective rank (ER [32, 18]) and downstream performance improved when training deep RL in the Atari domain, by defining hidden layers as Hadamard products [31]. Hadamard products between hidden layers enable richer high-dimensional interactions within the representation space, without increasing hidden layer dimensionality. This leads to more network capacity without explicitly scaling the network, which is often unstable in RL. Specifically, any hidden layer $z^j \in \mathcal{Z}$, with layer depth $j$, will be the Hadamard product of two parallel layers connected to the preceding hidden layer $z^{j-1}$:

$$z^j = f(z^{j-1}A_1^{j-1}) \odot f(z^{j-1}A_2^{j-1}), \tag{4}$$

where $A$ is a weight matrix, $f(*)$ is a nonlinear activation and the bias layers are left out for simplicity. As PQN employs layer normalization for training stability, and every representation is max-pooled, the final Hadamax representation layers can be defined as:

$$z^j = MP\Big(f\big(LN(z^{j-1}A_1^{j-1})\big) \odot f\big(LN(z^{j-1}A_2^{j-1})\big)\Big). \tag{5}$$

Where LN and MP represent layer normalization and max-pooling, respectively. It is worth noting that contrary to recent work on Hadamard representations [31], we show the possibility of successfully applying Hadamard products to zero-saturating activation functions such as ReLU or GELU [26]. We believe this is possible due to the relative training stability increase of PQN over DQN, as a result of applying LayerNorm and the removal of the target network and replay buffer. This training stability correlates with a minimal amount of dead neurons in the representation [51], which even leads to the ability to do element-wise multiplication of zero-saturating (sparse) neurons without increasing dead neurons.

## 4.3 Design Choice 3: Gaussian Error Linear Unit

The Gaussian Error Linear Unit (GELU) is used in various neural network architectures, the most notable applications being in transformer-based architectures such as BERT [12] and GPT [44].

It is defined as:
$$\text{GELU}(x) = x\Phi(x)$$

where $\Phi(x)$ is the cumulative distribution function of the standard normal distribution. Equivalently, it can be expressed using the error function as:

$$\text{GELU}(x) = 0.5x\left(1 + \text{erf}\left(\frac{x}{\sqrt{2}}\right)\right)$$

In contrast to the ReLU, which converts negative inputs to zero, GELU permits small negative values to pass through in a softened form (See Fig. 3), allowing more stable gradient flow for negative inputs. Overall, GELU has been shown to improve performance in various deep learning tasks, including computer vision and natural language processing [26]. In the Hadamax encoder, we therefore replace all the original ReLU activation functions with the GELU.

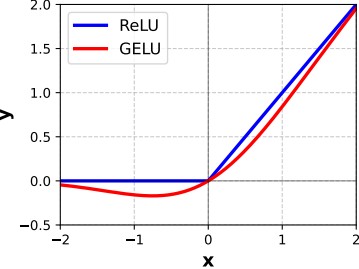

Figure 3: ReLU and GELU.

## 5 Experiments

We compare our agent against widely used model-free RL baselines across 57 Atari games. Through experiments, we aim to answer: (i) do agents equipped with Hadamax encoders outperform those using conventional encoders? (ii) what are the reasons behind Hadamax's superior performance? (iii) what is the impact of each proposed design choice?

**Baselines:** We compare our method with the following baselines: (1) DQN [37], a pioneer RL method that uses a deep neural network to play Atari, achieving human performance. (2) C51 [5],

Rainbow [27], a state-of-the-art model-free method, combining various algorithmic and architectural techniques together. (3) PQN [17], a recent novel parallel Q-learning network without a replay buffer and target network. In terms of performance, PQN is on par with C51, while remaining algorithmically less complex than DQN. Our final baseline is (4) PQN (ResNet-15), which combines PQN with the more complex Impala CNN architecture, used throughout modern state-of-the-art RL algorithms as a drop-in replacement for the conventional Nature encoder [13, 50].

**Environments:** The full 57-game Atari domain [6] is used as a standardized benchmark for evaluating our algorithm's performance. In line with best practices in the field, we focus on the median human-normalized score over all 57 games [38, 27, 21, 17]. To manage computational load, ablations are done on 40M frames, while comparison with baselines is done at the official 200M frame scores. Note that there can be relative differences between performances in 40M and 200M frames, as the epsilon-greedy coefficient $\epsilon$ is scaled

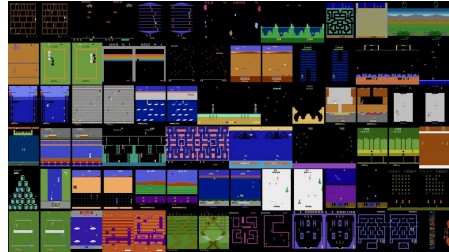

Figure 4: The Atari-57 domain.

down over the total training time. An algorithm seed initialized to run for 40M frames will therefore have a different convergence curve towards 40M than the same algorithm initialized for a 200M frames seed. We refer the reader to more detailed descriptions of environments and implementations of baseline agents in Appendix C.3.

## 5.1 Hadamax-PQN: Results

The full 200M frame training curves for PQN, PQN (ResNet-15) and Hadamax-PQN are shown in Fig. 5 (left). The Hadamax encoder clearly yields benefits over the widely used Impala ResNet-15 encoder [13], and causes PQN to significantly surpass Rainbow-DQN [27]. Although the original paper shows that PQN is able to beat Rainbow-DQN when training for around 260M environment frames [17], Hadamax-PQN reaches this score at around 90M frames. Another commonly used scoring method, the Atari-57 score profile, can be seen in Fig. 5 (right). Note that the scores used in this research for DDQN, C51 and Rainbow have been taken from the original papers, and are generally higher than their practical implementations on various GitHub repositories. For details on how to compute the median human-normalized score and the Atari score profile, we refer the reader to Appendix D.

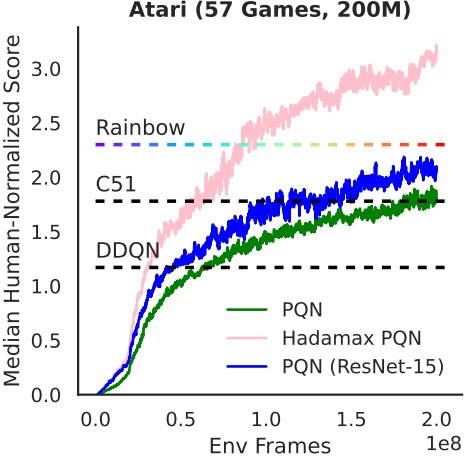
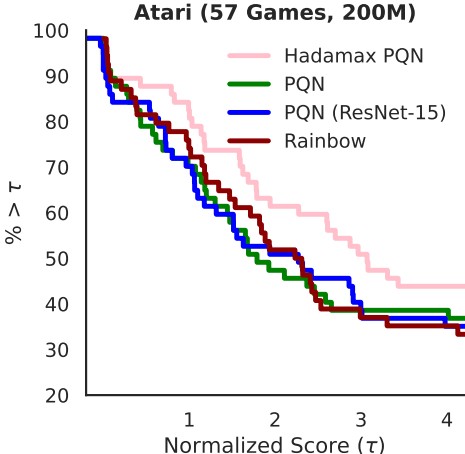

Figure 5: Median Human-Normalized performance training PQN, PQN (Resnet-15) and Hadamax-PQN in the Atari domain over 57 games, 200M frames and 5 seeds (left), and the Atari-57 score profile (right). The Atari-57 score profile illustrates the percentage of games that exceed the normalized score threshold on the x-axis.

The effect of the Hadamax encoder on the baseline PQN on a per-game basis can be seen in Fig. 6. The results show a significant performance increase over the baseline, with over 17 games having more than 100% improvement, compared to only one single game having more than a 50% decrease in performance. The per-game improvements over the Rainbow-DQN baseline can be seen in Appendix E.4. For each individual game's training curve and the final 200M frame score table, we refer the reader to Appendix F. To the best of our knowledge, the implementation of the Hadamax encoder is one of the biggest recorded non-algorithmic improvement over a recent competitive RL baseline, and it does not involve any complex hyper-parameter tuning.

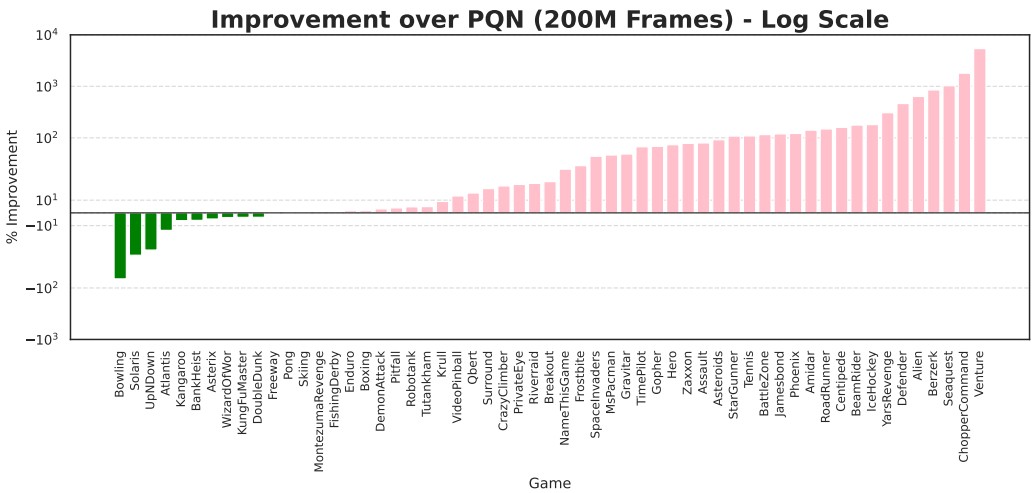

Figure 6: Per-game improvement of Hadamax-PQN over PQN (Log Scale).

## 5.2 Does Hadamax Generalize Beyond PQN?

The Hadamax encoder not only enhances the performance of PQN, but also works effectively with other reinforcement learning agents. To showcase this, the C51 algorithm is evaluated on the Atari-10 benchmark for 40M environment frames. As shown in Figure 7, a direct implementation of the Hadamax encoder to the C51 algorithm boosts the performance by approximately 70% on Atari-10 [2]. Similar to PQN, the algorithmic hyperparameters for Hadamax-C51 remain exactly the same as for the C51 baseline from *cleanrl* [28]. These improvements suggest that the Hadamax encoder is able to be implemented as a strong default encoder for multiple algorithms in the Atari domain. For more information on the Atari-10 benchmark and the corresponding score normalization metrics, we refer the reader to Appendix D.3.

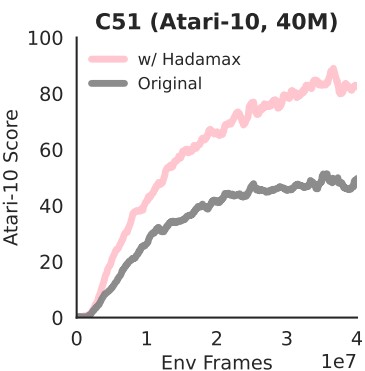

Figure 7: C51 with and without a Hadamax encoder on Atari-10.

## 5.3 Effective Rank and Dead Neurons

In order to obtain clues about the stabilizing effects of the proposed Hadamax encoder, the effective rank of the hidden layers is investigated during training [32, 18], as well as the amount of dead neurons [51]. The effective rank of a feature matrix for a threshold $\delta$ ($\delta = 0.01$), denoted as $srank_\delta(\Phi)$, is given by $srank_\delta(\Phi) = \min\left\{k : \frac{\sum_{i=1}^{k} \sigma_i(\Phi)}{\sum_{i=1}^{d} \sigma_i(\Phi)} \geq 1 - \delta\right\}$, where $\{\sigma_i(\Phi)\}$ are the singular values of $\Phi$ in decreasing order, i.e., $\sigma_1 \geq \cdots \geq \sigma_d \geq 0$. The effective rank portrays a measure of network capacity i.e. the amount of information that can be approximated in a certain hidden layer.

We investigate the differences in effective rank between the baseline PQN and Hadamax-PQN. To find clues for Hadamax's strong improvements on certain environments, the differences are visualized on a random subset of 5 high-improvement environments from Fig. 6. The effective rank of the

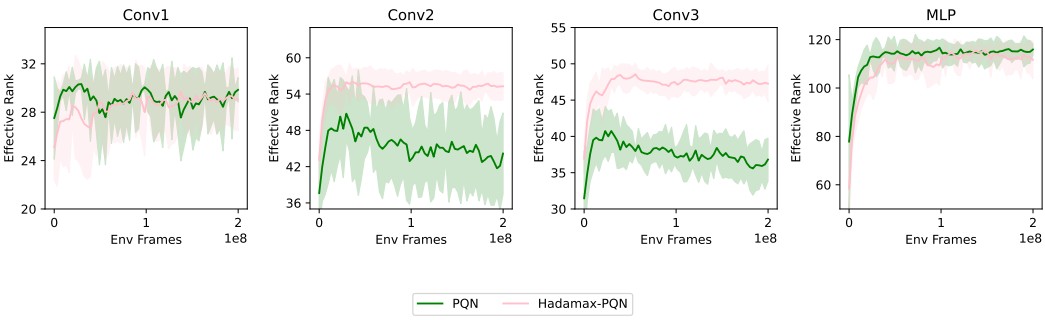

Figure 8: Effective rank [32] of the 4 hidden layers for both the baseline PQN and the Hadamax-PQN setting. Although there is no visible difference between the first and final layer, the deeper convolutional layers show a lower effective rank in the baseline setting, as well as a stronger rank decay during training.

encoder's representation layers while training for 200M frames can be seen in Fig. 8. The plots show that there are minimal differences in effective rank in the first and last hidden layer of the encoder. However, in the baseline PQN encoder, the deeper convolutional layers show a more prominent decay in rank during training, as well as a reduced initial effective rank. As mentioned in Section 4, the increase in effective rank in the deeper convolutional layers can largely be credited to the use of Hadamard representations. A further look is taken at the convolutional channel cosine similarity [54], where a low cosine similarity indicates that the channels are extracting dissimilar or uncorrelated features from the input data, which is desirable and suggests diversity among the channels. An ablation of the max-pool and Hadamard components' effect on both effective rank and channel cosine similarity can be seen in Table 1. The Hadamax encoder improves both the effective rank and channel cosine similarity, when compared to the baseline encoder. This indicates that Hadamax extracts more expressive, uncorrelated features from the pixel inputs.

Table 1: Channel Cosine Similarity & Effective Rank

| Metric | Baseline | + Maxpool | + Hadamard | Both (Hadamax) |
|---|---|---|---|---|
| Effective Rank | Base | +10% | +10% | +10-20% |
| Channel Cosine Similarity | Base | +20% | -90% | -50% |

Further investigation into the penultimate layer's fraction of dead neurons shows a small decrease from the baseline (see Fig. 9). The percentage of dead neurons in the final hidden layer is calculated by finding neurons that have a variance of less than $10^{-4}$ over the batch dimension. In practice, this metric generalizes well to any activation function (ReLU, GELU, Tanh). After training for 200M frames, both the baseline PQN and Hadamax-PQN have less than 8% dead neurons, which remains extremely low compared to DQN [51]. We therefore do not expect a substantial correlation between the small reduction in dead neurons and the performance. However, in contrast to recent work on Hadamard representations [31], who showed that the DQN algorithm exhibits instability when multiplying ReLU-activated neurons, we show that it is possible to use Hadamard products on zero-saturating activations. We believe the inherent stability of the PQN algorithm and its corresponding low fraction of dead neurons allows for successful Hadamard multiplication of linear-unit activations like ReLU or GELU.

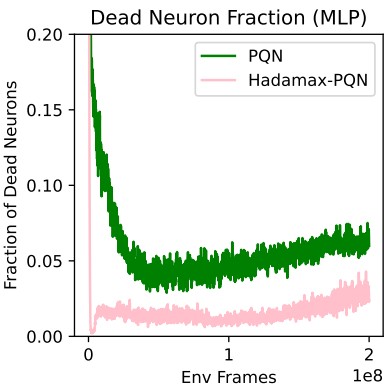

Figure 9: Fraction of dead neurons over 200M frames.

## 5.4 Which Design Choice is most Important?

As described in Section 4, the Hadamax encoder differs from PQN's conventional Nature CNN encoder in three areas: (1) applying max-pooling (2) using Hadamard representations and (3) GELU-activated hidden layers. The precise influence of each component of the Hadamax encoder remains to be determined. An ablation analysis over these areas is therefore done on 40M environment frames in the full Atari-57 suite. The ablations are defined as implementation subtractions from the original Hadamax architecture in Fig. 2. The result of the ablation study is shown in Fig. 10a. Next to the ablations, the effects of direct additions of our design choices on the baseline PQN are investigated. The results of the addition analysis can be seen in Fig. 10b.

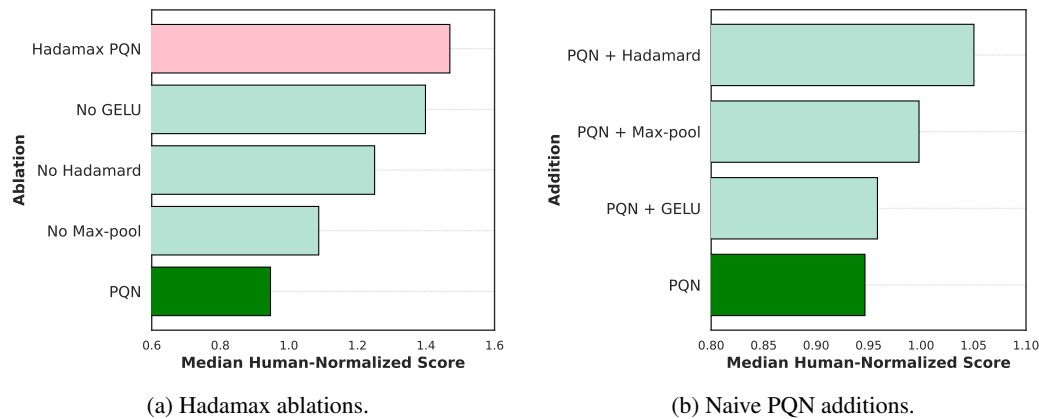

(a) Hadamax ablations.  (b) Naive PQN additions.

Figure 10: Ablations of Hadamax-PQN, each represented as a subtraction from the full Hadamax architecture (a), and naive architectural additions to the baseline PQN (b).

Over a training period of 40M frames, the subtraction of max-pooling leads to the largest decay in performance. Note that when max-pooling is subtracted from our architecture, we return the convolutional strides to their original values, in order to still retain feature compression. The importance of down-sampling with max-pooling strengthens our hypothesis that a selection of the most prominent features is key when working with high-dimensional observation spaces in the Atari domain. The use of convolutional Hadamard representations is also an important component, showing that the increase in effective rank paired with other benefits such as high-order interactions [11], have a strong correlation with downstream performance. Finally, the GELU activation has the lowest importance, although its contribution as compared to the ReLU still remains substantial for such a small architectural component. Notably, if the ablations are compared to the effects of directly implementing a single design choice on the baseline (see Fig. 10b), it becomes clear that the overall combination of all three components is a key factor. For an experimental analysis with two deeper Hadamax encoders, we refer the reader to Appendix E.2.

## 5.5 VizDoom

Additional experiments have been conducted on the pixel-based VizDoom environment [30]. Training RL on VizDoom is different from Atari as it works with 3D environments, semi-realistic physics, and stochastic elements demanding more advanced reasoning. On VizDoom Deathmatch, the baseline convolutional encoder was converted to a Hadamax encoder, following the same shallow convolutional architecture with filter sizes of 32-64-128. The RL baseline used is the actor-critic PPO algorithm [43, 48]. Without changing any other hyperparameters, the results after training for 40M frames can be seen in Table 2.

Table 2: VizDoom Deathmatch performance

| Method | 10M | 20M | 30M | 40M |
|---|---|---|---|---|
| Hadamax | $-1.62 \pm 0.20$ | $\mathbf{8.08} \pm 1.61$ | $\mathbf{19.36} \pm 1.82$ | $\mathbf{29.38} \pm 10.31$ |
| Baseline | $-2.81 \pm 0.40$ | $1.27 \pm 0.80$ | $3.10 \pm 0.89$ | $5.21 \pm 1.27$ |

The strong performance increase over the baseline encoder suggests that Hadamax is applicable on actor-critic architectures, as well as a wider variety of pixel-based environments.

# 6 Conclusions and Future Work

This paper introduced the Hadamax encoder architecture, augmenting the conventional pixel-based Nature CNN architecture with **Hada**mard representations, while down-sampling using **max**-pooling instead of convolutional strides. Furthermore, the Gaussian Error Linear Unit activation was implemented to improve training stability. The application of these fundamental changes to the PQN baseline encoder, while preserving its original shallow structure, allowed for a profound increase in performance over several model-free baselines. Specifically, we reach an almost two-fold performance gain over the baseline PQN setting, and surpass Rainbow-DQN's official 200M frame score after just 90M frames, while remaining an order of magnitude faster. Additional results on C51 and PPO/VizDoom show that the Hadamax encoder remains effective across a variety of algorithms and across other pixel-based environments.

Due to computational constraints, this paper includes only limited testing of the Hadamax encoder on complex algorithms such as Bigger-Better-Faster (BBF) [50] on the Atari-100k benchmark or on a state-of-the-art model-based algorithm such as Dreamerv1-v3 [22, 23]. However, as seen by the performance improvement on C51 in Fig. 7 and PPO/Vizdoom in Table 2, we do expect a certain degree of generalization across algorithms and/or environments. Another limitation is that the Hadamax encoder, due to its increased architectural complexity, accounts for some extra computational overhead compared to PQN's conventional Nature CNN architecture. For completeness, the training durations are therefore reported in Fig. 1 and the inference durations are reported in Table 3. Inference time is however usually not a key issue in the RL context.

Table 3: Inference times

| Architecture | Inference time (milliseconds) |
|---|---|
| Rainbow | 0.59 |
| PQN | 0.39 |
| PQN (Impala) | 1.40 |
| Hadamax-PQN | 1.75 |

All in all, we believe this paper takes an important step forward in functional encoder synthesis for RL, discovering an alternative for the usual deep and complex ResNet architectures to optimize performance. An interesting avenue for future work would be to investigate scaling of the Hadamax encoder, as it already achieves significant performance improvements using only 3 convolutional layers and the classic 32-64-64 filter dimensions. Finding successful ways to scale the Hadamax encoder in either width or depth could yield even stronger improvements and more insights into architecture synthesis. Another promising avenue would be to explore the integration of MoE-style prediction heads in the Hadamax encoder, since common implementations of MoE do not necessarily affect the base encoder [42]. Furthermore, as Hadamax-PQN does not come with any algorithmic or hyperparameter changes, it can be used as a new baseline to build other algorithmic improvements upon. Specifically, since hard-exploration games are generally not suited for PQN's epsilon-greedy exploration regime, augmenting PQN-Hadamax with novel exploration techniques might further bridge the gap in performance between compute-light model-free and compute-heavy model-based algorithms such as DreamerV3 or Muzero [23, 47].

## Acknowledgements

We would like to thank Prof. Mark Hoogendoorn for his helpful guidance during this project. We also thank SURF (www.surf.nl) for the support in using the National Supercomputer Snellius. This work used the Dutch national e-infrastructure with the support of the SURF Cooperative using grant no. EINF-13858.

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

# Appendix

## Table of Contents

# A Impact Statement

This work shows that architectural innovations like the Hadamax encoder can drive significant progress in reinforcement learning. By enabling more efficient and accessible AI, it encourages broader adoption and exploration of learning systems across diverse real-world domains.

# B Hadamax Encoder Code

We provide the full JAX-based code of the Hadamax encoder for reproducibility purposes.

```python
# Input = input_obs, a frame-stacked Atari observation
x = jnp.transpose(input_obs, (0, 2, 3, 1))
x = x / 255.0
# First block
x1 = nn.Conv(32, kernel_size=(8, 8), strides=(1, 1), padding="SAME",
             kernel_init=nn.initializers.xavier_normal())(x)
x2 = nn.Conv(32, kernel_size=(8, 8), strides=(1, 1), padding="SAME",
             kernel_init=nn.initializers.xavier_normal())(x)
x1 = normalize(x1)  # Normalize before activation
x2 = normalize(x2)  # Normalize before activation
x1 = nn.gelu(x1) # Apply activation
x2 = nn.gelu(x2) # Apply activation
x = x1 * x2  # Hadamard product
x = max_pool(x, window_shape=(4, 4), strides=(4, 4), padding="SAME")
# Second block
x1 = nn.Conv(64, kernel_size=(4, 4), strides=(1, 1), padding="SAME",
             kernel_init=nn.initializers.xavier_normal())(x)
x2 = nn.Conv(64, kernel_size=(4, 4), strides=(1, 1), padding="SAME",
             kernel_init=nn.initializers.xavier_normal())(x)
x1 = normalize(x1)  # Normalize before activation
x2 = normalize(x2)  # Normalize before activation
x1 = nn.gelu(x1)  # Apply activation
x2 = nn.gelu(x2)  # Apply activation
x = x1 * x2  # Hadamard product
x = max_pool(x, window_shape=(2, 2), strides=(2, 2), padding="SAME")
# Third block
x1 = nn.Conv(64, kernel_size=(3, 3), strides=(1, 1), padding="SAME",
             kernel_init=nn.initializers.xavier_normal())(x)
x2 = nn.Conv(64, kernel_size=(3, 3), strides=(1, 1), padding="SAME",
             kernel_init=nn.initializers.xavier_normal())(x)
x1 = normalize(x1)  # Normalize before activation
x2 = normalize(x2)  # Normalize before activation
x1 = nn.gelu(x1)  # Apply activation
x2 = nn.gelu(x2)  # Apply activation
x = x1 * x2  # Hadamard product
x = max_pool(x, window_shape=(3, 3), strides=(1, 1), padding="SAME")
# Flatten for MLP layer
x = x.reshape((x.shape[0], -1))
x = nn.Dense(512, kernel_init=nn.initializers.he_normal())(x)
x = normalize(x)
x = nn.gelu(x)
x = nn.Dense(self.action_dim, name="action_dense")(x) # Final Q-Values
```

# C Experiment Details

## C.1 Hyperparameters

Table 4: Atari Hyperparameters for PQN, PQN (ResNet-15) and Hadamax-PQN. These hyperparameters are equal to the original hyperparameters from the PQN baseline [17].

| Parameter | Value |
|---|---|
| NUM_ENVs | 128 |
| NUM_STEPS | 32 |
| EPS_START | 1.0 |
| EPS_FINISH | 0.001 |
| EPS_DECAY | 0.1 |
| NUM_EPOCHS | 2 |
| NUM_MINIBATCHES | 32 |
| NORM_INPUT | False |
| NORM_TYPE | layer_norm |
| LR | 0.00025 |
| MAX_GRAD_NORM | 10 |
| LR_LINEAR_DECAY | False |
| GAMMA | 0.99 |
| LAMBDA | 0.65 |
| OPTIMIZER | RAdam |

## C.2 Environments

We run experiments on the Atari-57 suite, where there are 57 different games in total. No per-game tuning is allowed and the same agent architecture, hyper-parameters and pre-processing needs to run on every game. The suite contains varying games that can be used to examine different properties of RL agents, e.g. long-horizon credit assignment, partial observability, hard exploration, etc.

Each observation consists of 4 grayscale images of the game state stacked together, i.e. $(4, 64, 64)$. The action space is discrete, and each action represents a different operation in the game. The reward function depends on the environment chosen. More details on each game can be found at `https://ale.farama.org`.

**Atari-3 and Atari-10**: We examine C51, DQN and Rainbow on Atari-3 or Atari-10 [2], which are a small but representative subset of the full Atari-57 suite. Atari-3 includes Battle Zone, Name This Game and Phoenix. Atari-10 includes Amidar, Bowling, Frostbite, Kung Fu Master, River Raid, Battle Zone, Double Dunk, Name This Game, Phoenix and Q*Bert.

## C.3 Baseline Implementations

**PQN**: We use the official codebase [2] of PQN and default hyper-parameter settings.

**Rainbow, C51, DQN**:For the Fig. 12 training results we use implementations from *cleanrl* [3] and default hyper-parameter settings. The scores for DDQN, C51 and Rainbow in figures 1 and 5 have been taken from their respective official papers.

**Hadamax encoder**: Since the whole PQN codebase is in Jax, we implement the Hadamax encoder for PQN in Jax as well. As Implementations of Rainbow, C51 and DQN from *cleanrl* are in PyTorch, we also implement the Hadamax encoder for these agents in PyTorch.

## C.4 Compute Usage

We run all our experiments on a HPC cluster equipped with A100 GPUs. Each run of Hadamax-PQN needs around 45 minutes for 40 millions frames and PQN needs around 20 minutes.

---

[2] `https://github.com/mttga/purejaxql`
[3] `https://github.com/vwxyzjn/cleanrl`

# D Metrics

## D.1 Median Human-Normalized Score

For each game, compute the average score $x_i$ across multiple independent seeds. Then compute the normalized score $Z_i$ as:

$$Z_i = \frac{x_i - r_i}{h_i - r_i}$$

where $x_i$ is the raw score, and $r_i$ and $h_i$ are the random and human scores for game $i$, respectively (see Table 6 for values). After computing the normalized scores for all 57 games * seeds, they are sorted and the median value is computed.

## D.2 Atari-57 Score Profile

**x-axis**:($\tau$ - Normalized Score). Represents the threshold score (e.g., Human-Normalized Score). Higher values mean better performance.

**y-axis**: $\tau\%$ = fraction of games above $\tau$. Shows the fraction of games for which the agent's normalized score is greater than $\tau$. For example, at $\tau = 1$, the y-value represents what fraction of games the agent beats $\tau = 1$ human performance on. In other words, it represents the percentage of games that has scores higher than $\tau$.

## D.3 Atari-3 and Atari-10

The Atari-3 and Atari-10 scores approximate the median normalized score across the full 57-game Atari benchmark using subsets of 3 and 10 games, respectively [2]. The computation involves the following steps:

1. For each game in the subset, compute the normalized score $Z_i$ as:

$$Z_i = 100 \times \frac{x_i - r_i}{h_i - r_i}$$

   where $x_i$ is the raw score, and $r_i$ and $h_i$ are the random and human scores for game $i$, respectively (see Table 6 for values).

2. Apply the log transform:

$$\phi(Z_i) = \log_{10}(1 + \max(0, Z_i))$$

3. Compute the weighted sum $f = \sum_{i \in I} c_i \phi(Z_i)$, where $I$ is the subset of games and $c_i$ are the subset-specific coefficients.

4. Obtain the predicted median score as:

$$\hat{t} = 10^f - 1$$

For Atari-3, the subset comprises Battle Zone, Name This Game, and Phoenix, with coefficients $c_i = [0.3706, 0.5133, 0.1015]$.

For Atari-10, the subset includes Amidar, Bowling, Frostbite, Kung Fu Master, River Raid, Battle Zone, Double Dunk, Name This Game, Phoenix, and Q*Bert, with coefficients $c_i = [0.0825, 0.0559, 0.0691, 0.0986, 0.0486, 0.1888, 0.0852, 0.1287, 0.1643, 0.0592]$.

# E   Additional Experiments

## E.1   Memory Usage

Table 5: Memory usage and batch sizes for different architectures

| Architecture | Training Update Memory (MB) | Batch | Inference Memory (MB) | Inference Batch |
|---|---|---|---|---|
| Rainbow | 198.77 | 32 | 185.33 | 1 |
| PQN | 254.54 | 256 | 155.87 | 8 |
| PQN (Impala) | 725.18 | 256 | 143.46 | 8 |
| Hadamax-PQN | 2247.26 | 256 | 233.14 | 8 |

## E.2   Deeper Hadamax Networks

As network scaling has become a topic of interest in the field of RL [35, 50, 42], we provide experiments using deeper versions of our encoder: 5-layer and 7-layer Hadamax-PQN. Specifically, the second and third convolutional layers in the original 3-layer encoder are duplicated, and we refrain from max-pooling the duplicates to avoid excessive compression. Similar to the ablations, the deep networks are tested on the full 57-game Atari suite for 40M environment frames. The results can be seen in Fig. 11.

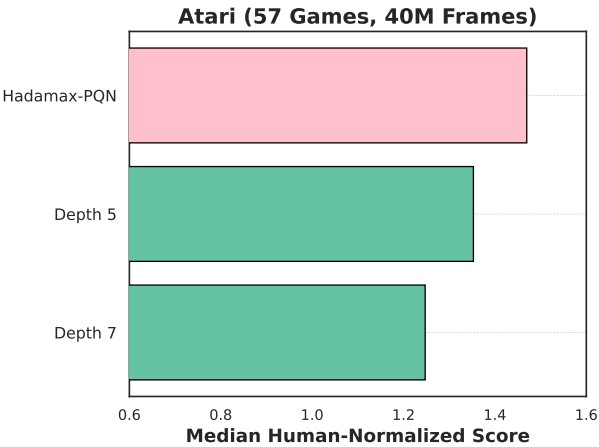

Figure 11: Hadamax encoder depth Ablations.

Simply using deeper convolutional Hadamax encoders does not seem to improve performance. Although there are more promising ways to scale the Hadamax encoder both in depth and width, the computational cost was the limiting factor in pursuing this in more detail. As discussed in the main paper, we leave this as a promising research area for future work.

### E.3 Hadamax with Other Agents

We modify the encoders of the widely-used *cleanrl* [28] implementations of C51, DQN, and Rainbow to demonstrate that the Hadamax encoder can generalize across various model-free agents. See Figure 12, on Atari-10, Hadamax improves the performance of the original C51 by 70%, and on Atari-3, it boosts DQN and Rainbow by 20% and 30%, respectively. These substantial gains, achieved by simply replacing the encoder, suggest that Hadamax could serve as a new default encoder for model-free reinforcement learning methods on Atari.

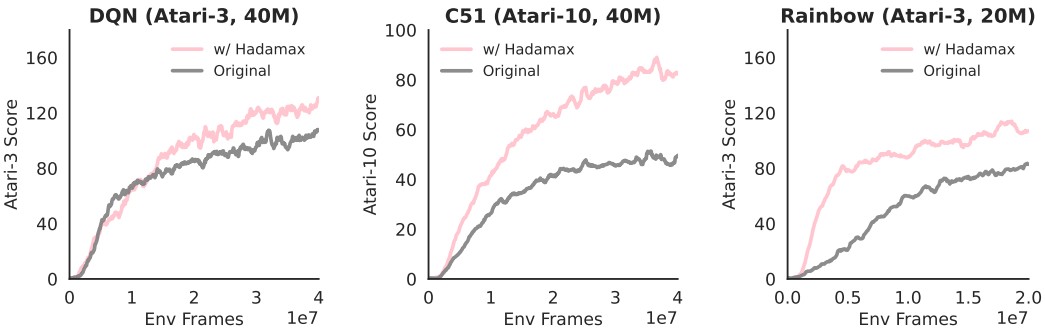

Figure 12: Performance gains of DQN, C51 and Rainbow with Hadamax encoders on a subset of Atari-57.

### E.4 Per-game improvement over Rainbow-DQN

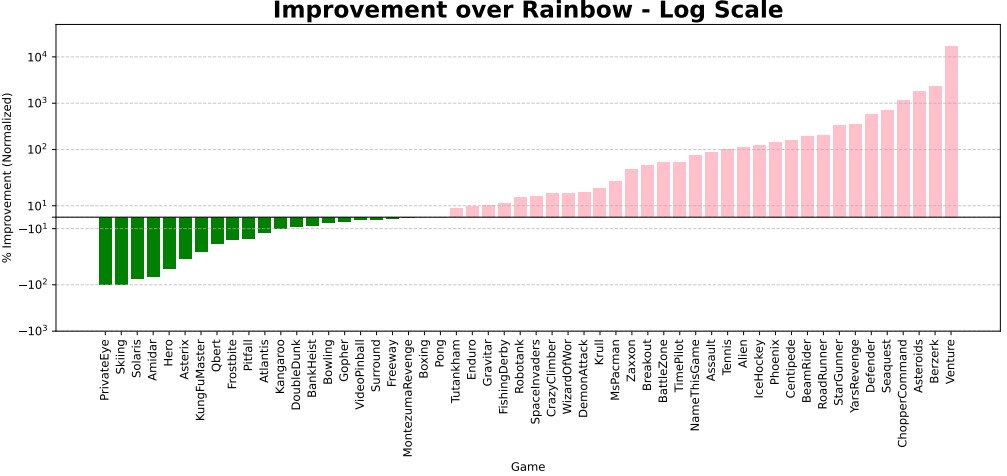

# F   Individual Game Scores

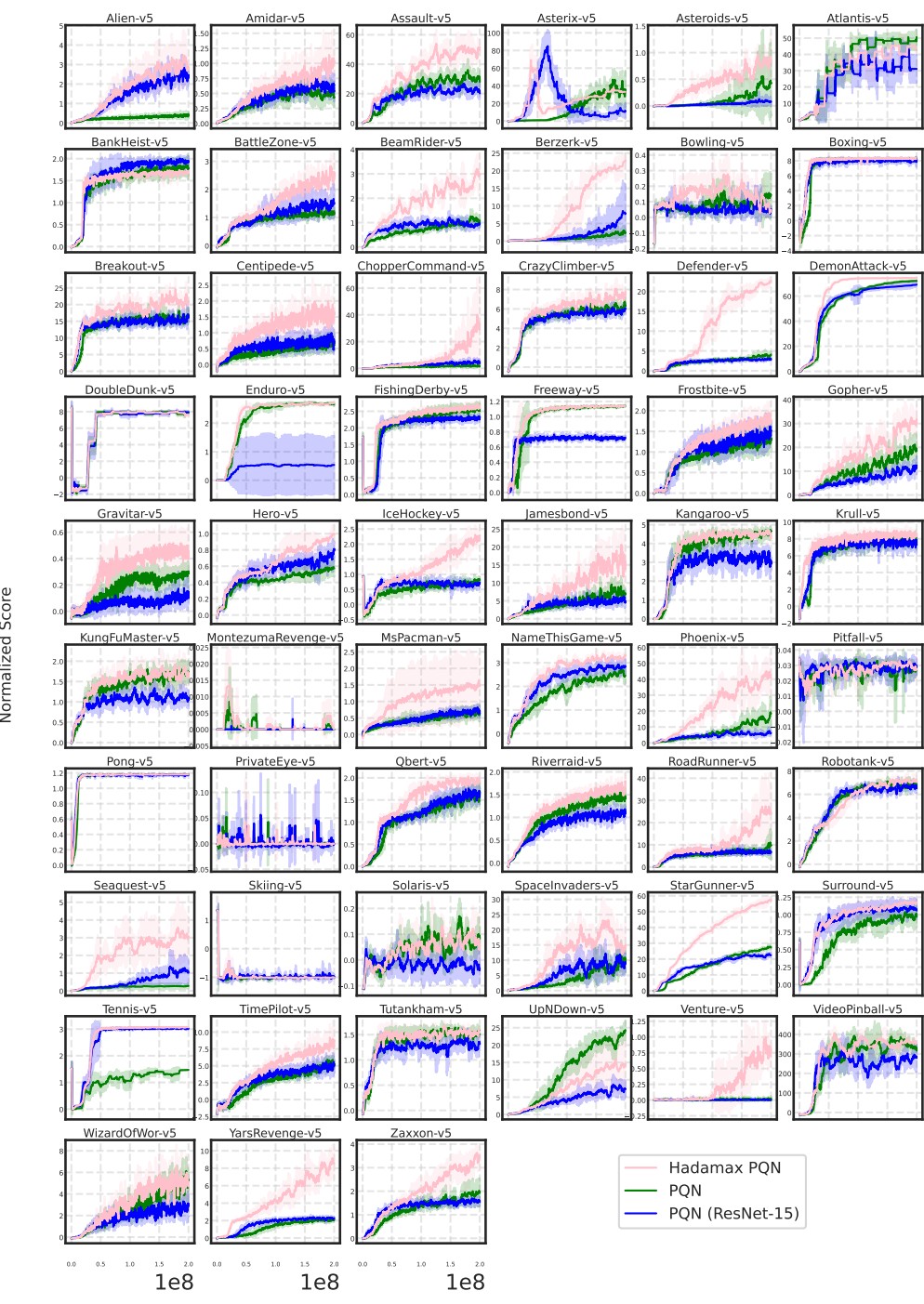

Table 6: Final 200M frame scores.

| Game | Hadamax-PQN | PQN (Resnet-15) | PQN |
|---|---|---|---|
| Alien-v5 | **20045.4** | 16935.0 | 2916.3 |
| Amidar-v5 | **1774.5** | 944.1 | 740.3 |
| Assault-v5 | **26426.7** | 11160.5 | 15089.7 |
| Asterix-v5 | 274915.0 | 95400.0 | **287697.6** |
| Asteroids-v5 | **39328.0** | 4232.7 | 21047.6 |
| Atlantis-v5 | 715750.0 | 516357.8 | **831884.7** |
| BankHeist-v5 | 1260.2 | **1446.1** | 1336.2 |
| BattleZone-v5 | **92951.2** | 55106.5 | 44130.8 |
| BeamRider-v5 | **49480.9** | 16315.8 | 18131.7 |
| Berzerk-v5 | **57497.4** | 19597.8 | 6061.3 |
| Bowling-v5 | 29.7 | 30.2 | **42.5** |
| Boxing-v5 | **99.8** | 96.3 | 98.3 |
| Breakout-v5 | **607.4** | 470.3 | 489.6 |
| Centipede-v5 | **17901.7** | 9266.5 | 8178.2 |
| ChopperCommand-v5 | **203593.5** | 26974.7 | 11688.8 |
| CrazyClimber-v5 | **202048.2** | 162776.7 | 168732.4 |
| Defender-v5 | **360287.5** | 48761.1 | 66381.0 |
| DemonAttack-v5 | **135450.5** | 125870.4 | 131320.0 |
| DoubleDunk-v5 | -1.8 | **-1.2** | **-1.2** |
| Enduro-v5 | **2323.4** | 462.7 | 2284.6 |
| FishingDerby-v5 | **46.6** | 31.2 | 45.4 |
| Freeway-v5 | **33.7** | 21.4 | **33.8** |
| Frostbite-v5 | **7689.5** | 6537.2 | 5623.8 |
| Gopher-v5 | **67829.0** | 26859.5 | 40834.5 |
| Gravitar-v5 | **1547.2** | 514.4 | 1107.3 |
| Hero-v5 | **30617.4** | 24912.9 | 18099.9 |
| IceHockey-v5 | **16.3** | -2.4 | -1.4 |
| Jamesbond-v5 | **4244.8** | 1285.8 | 1942.8 |
| Kangaroo-v5 | 13177.3 | 8728.6 | **13992.5** |
| Krull-v5 | **10554.4** | 9497.9 | 9802.2 |
| KungFuMaster-v5 | 36751.9 | 24102.8 | **38233.3** |
| MontezumaRevenge-v5 | **0.0** | **0.0** | **0.0** |
| MsPacman-v5 | **6968.3** | 4584.7 | 4909.7 |
| NameThisGame-v5 | **21334.2** | 18754.3 | 16437.0 |
| Phoenix-v5 | **267080.2** | 41001.4 | 120959.5 |
| Pitfall-v5 | -43.5 | **-34.4** | -50.5 |
| Pong-v5 | **21.0** | 20.8 | **21.0** |
| PrivateEye-v5 | 3.6 | 3.9 | **7.5** |
| Qbert-v5 | **25970.2** | 21818.4 | 22449.6 |
| Riverraid-v5 | **29423.9** | 18669.8 | 24133.3 |
| RoadRunner-v5 | **190019.6** | 52925.0 | 76600.9 |
| Robotank-v5 | **71.5** | 66.1 | 68.3 |
| Seaquest-v5 | **129408.8** | 43559.8 | 11554.4 |
| Skiing-v5 | -29971.3 | **-29479.8** | -29972.3 |
| Solaris-v5 | 1884.2 | 863.5 | **2189.4** |
| SpaceInvaders-v5 | **22258.0** | 13800.3 | 15125.0 |
| StarGunner-v5 | **549350.4** | 215397.7 | 264413.1 |
| Surround-v5 | **9.4** | 7.5 | 6.3 |
| Tennis-v5 | **23.8** | 22.9 | -1.0 |
| TimePilot-v5 | **17946.0** | 11924.0 | 12320.1 |
| Tutankham-v5 | **258.7** | 216.8 | 248.0 |
| UpNDown-v5 | 191857.2 | 82743.3 | **270833.7** |
| Venture-v5 | **940.7** | 0.0 | 18.1 |
| VideoPinball-v5 | **522510.3** | 416690.8 | 463022.1 |
| WizardOfWor-v5 | 21526.1 | 13130.5 | **22214.2** |
| YarsRevenge-v5 | **444710.8** | 119951.1 | 111611.7 |
| Zaxxon-v5 | **31400.8** | 14229.4 | 17644.0 |

