# OpenReview forum: "Hadamax Encoding: Elevating Performance in Model-Free Atari"
_NeurIPS.cc/2025/Conference — NeurIPS 2025 poster_

### Official Review · Reviewer_kiFq · 2025-07-02

**Clarity:** 4
**Significance:** 3
**Originality:** 3
**Rating:** 4
**Confidence:** 4

**Summary:**

This paper introduces the Hadamax encoder, a pixel-based neural network architecture for model-free reinforcement learning that combines max-pooling down-sampling, Hadamard products between parallel hidden layers, and GELU activations. Integrated into the Policy Quantization Network (PQN), Hadamax raises median human-normalized scores on Atari-57 by roughly 80% over vanilla PQN, surpasses Rainbow-DQN while training more than an order of magnitude faster, and transfers well to C51.

**Questions:**

- An exploration of how Hadamax integrates with data augmentation techniques such as DrQ, as well as with pruned or mixture-of-experts encoders, could help determine the compatibility of the proposed architecture with modern modular approaches.
- It would be helpful to understand the parameter allocation between the encoder and the policy/value head, and how this distribution compares with other architectural configurations.
- An evaluation of Hadamax on continuous-control benchmarks, such as DMC or Adroit, could offer insight into its generalization capabilities in domains requiring fine-grained spatial reasoning.
- Clarification on the exclusion of residual connections and deeper networks would be appreciated, along with discussion on whether such design choices were influenced by concerns over diminishing returns or optimization instability.
- Investigating the effects of substituting max-pooling with strided convolutions while preserving Hadamard interactions—or vice versa—could further isolate the contribution of each architectural component.
- A quantitative analysis of GPU memory consumption and inference throughput in comparison to Rainbow-DQN would provide a more complete picture of the computational trade-offs introduced by the Hadamard operations.

**Ethical Concerns:**

["NO or VERY MINOR ethics concerns only"]

**Final Justification:**

I have reviewed the comments given by the reviewers and find that many of my concerns have been resolved and thus maintain my positive score.

**Limitations:**

yes

**Quality:**

3

**Strengths And Weaknesses:**

Strengths

- The reported 80% performance gain over the PQN baseline and the outperformance of Rainbow-DQN with >10× shorter training times demonstrate clear practical impact.
- The encoder’s design is conceptually simple—replacing strides with max-pooling and inserting an element-wise Hadamard branch—so practitioners can drop it into existing codebases with minimal engineering effort, but enjoy massive performance gains.
- The ablation study cleanly isolates the contributions of max-pooling, Hadamard products, and GELU activations, strengthening the causal link between the architectural decisions and the observed gains.

Weaknesses
- All experiments are confined to the ALE 200M benchmark, leaving open whether the encoder generalizes to domains that demand fine-grained spatial reasoning or continuous actions, such as robotics or locomotion related environments such as DMC, Maniskill and Adroit.
- The study compares mainly to PQN, an algorithmic rather than architectural baseline, and omits direct comparison to architectural strategies such as SoftMoE [1] and network pruning [2] based approaches.
- The architecture exploration is shallow: residual connections are excluded, depth is limited, and modern components such as attention or normalization variants are not considered.
- Computational analysis focuses on wall-clock training time but omits memory usage and inference latency.

[1] Mixtures of experts unlock parameter scaling for deep rl, Obando-Ceron et al., ICML 2024
[2] In value-based deep reinforcement learning, a pruned network is a good network, Obando-Ceron et al., ICML 2024

---

> ### Author Rebuttal · Authors · 2025-07-31
>
> We thank Reviewer kiFq for the extensive review, and acknowledging the practical impact of our encoder design. As the Reviewer’s stated weaknesses are similar to the questions, we will stick to addressing the questions 1 by 1.
>
> - We have done tests integrating Hadamax and the baseline with a mixture-of-expert encoder head, specifically the SoftMoE head [1]. Using a SoftMoE head, we noticed that  the performance dropped. We were hesitant to report this, as we could have made an implementation error, which would not have been fair to the SoftMoE authors. Furthermore, the authors of SoftMoE and RL network pruning [2] are working with, to the best of our knowledge, a different Atari domain with sticky-actions, and mostly experiment on a subset of the 57-game suite. It is therefore not possible to do objective comparisons using scores from their papers. In order for us to compare to the official results in the Rainbow and PQN paper, we follow the 57-game, 200M frame implementation on Atari without sticky actions, and report the median Human-normalized scores. Having said this, we do see strong potential in future work integrating e.g. SoftMoE with Hadamax, as Hadamax is focused on the convolutional part of the encoder, while the Mixture of Expert models are focused on the bottleneck part of the encoder.
> - We have calculated the parameter allocation between the encoder and the policy/value head:
>
> | Architecture | Encoder Parameters | Head Parameters | Encoder % of Total | Head % of Total | Total Parameters |
> |--------------|--------------------|-----------------|--------------------|-----------------|------------------|
> | Hadamax | 4,123,072 | 9,234 | 99.78% | 0.22% | 4,132,306 |
> | Baseline | 1,685,472 | 9,234 | 99.45% | 0.55% | 1,694,706 |
> | Rainbow | 77,984 | 7,418,770 | 1.04% | 98.96% | 7,496,754 |
> | Impala | 596,312 | 4,626 | 99.23% | 0.77% | 600,938 |
>
> - We agree that a Hadamax implementation on continuous control benchmarks would offer more insights. However, pixel-based continuous control is, like Atari-57, very compute intensive, as it requires a lot of samples to converge as opposed to state-based continuous control. Even though we concur that testing the capabilities in fine-grained spatial reason would be very interesting, we respectfully leave the integration of the Hadamax encoder in the pixel-based continuous control domain for future work. We did however run an additional evaluation experiment on the actor-critic architecture PPO in Vizdoom (Deathmatch), where we replaced the baseline encoder with Hadamax. While not continuous control, VizDoom also demands spatial reasoning in a dynamic FPS environment. Hadamax also achieves strong performance benefits over the baseline encoder in VizDoom, as shown in the following table:
>
> | VizDoom Deathmatch | 10M | 20M | 30M | 40M |
> |---------|--------------|-------------|--------------|---------------|
> | Hadamax | **-1.62** ± 0.20 | **8.08** ± 1.61 | **19.36** ± 1.82 | **29.38** ± 10.31 |
> | Baseline | -2.81 ± 0.40 | 1.27 ± 0.80 | 3.10 ± 0.89 | 5.21 ± 1.27 |
>
>
> - In Appendix E.1 we have an ablation where we ‘naively’ scale Hadamax networks in depth, which shows that simply scaling in depth yields decreasing returns for Hadamax (and for the baseline, but that is not shown here). Regarding the residual connections, IMPALA's architecture separates channel increases and downsampling into initial convolutions and pooling layers within the Conv sequences, allowing the residual blocks to operate on constant channels and spatial sizes for identity skips without needing projections. Our Hadamax encoder does not have this, creating channel mismatches that prevent direct residuals. We therefore did not experiment with residual connections. If you would prefer, we are willing to add an ablation to the camera-ready showing (projected) residual connections on the Hadamax encoder.
> - Sorry for the confusion, but do you mean, taking the full Hadamax encoder, but substituting the max-pooling with strided convolutional layers, thereby increasing the encoder’s convolutional depth? If that is the case, we are also able to add this ablation to the camera-ready version.
> - We have tested the inference times on the Pytorch models:
>
> | Architecture | Inference time (seconds) |
> |----------------|----------------|
> | Rainbow | 0.00059 |
> | PQN | 0.00039 |
> | PQN (Impala) | 0.00140 |
> | Hadamax-PQN | 0.00175 |
>
> And also the GPU memory of the Pytorch models:
>
> | Architecture   | Training Update Memory (MB) | Training Batch | Inference Memory (MB) | Inference Batch |
> |-----------------|----------------------|---------------------|----------------|----------------------|
> | Rainbow | 198.77          	| 32             	| 185.33    	| 1               	|
> | PQN| 254.54          	| 256            	| 155.87    	| 8               	|
> |PQN (Impala)  | 725.18          	| 256            	| 143.46    	| 8               	|
> | Hadamax-PQN | 2247.26         	| 256            	| 233.14    	| 8               	|
>
>
> These tests were done on an Nvidia 3080 Laptop GPU. As expected, Hadamax-PQN uses more GPU memory than Rainbow. However, we kindly emphasize that these numbers are still only a fraction of the available GPU memory on any recent GPU. Furthermore, as also stated in the original PQN paper, PQN saves 26GB of replay-buffer RAM compared to Rainbow.
>
> We thank you for your detailed review, and will implement the additional clarifications / ablations in the main paper and appendices of the camera-ready version, as they will certainly strengthen the paper.
>
> [1] Mixtures of experts unlock parameter scaling for deep rl, Obando-Ceron et al. 2024.
>
> [2] In value-based deep reinforcement learning, a pruned network is a good network, Obando-Ceron et al, 2024.

---

> > ### Comment · Reviewer_kiFq · 2025-08-07
> >
> > Thank you for the thoughtful and constructive responses. I appreciate the clarifications provided, which have addressed most of my concerns.
> >
> > Regarding the comment on substituting max-pooling with strided convolution layers—yes, that was indeed my suggestion. My intention was to understand the impact of max-pooling, as downsampling could also be achieved through stacked convolutional layers. However, I acknowledge that this is more of an exploratory direction rather than a required experiment.
> >
> > That said, I believe my original evaluation fairly reflects the paper’s current strengths and limitations. I would therefore prefer to maintain my score of Borderline Accept.

---

### Official Review · Reviewer_PtqA · 2025-07-02

**Clarity:** 4
**Significance:** 2
**Originality:** 2
**Rating:** 4
**Confidence:** 4

**Summary:**

In this paper, the authors focus on the problem of model free reinforcement learning applied to the Atari video game domain. To this end, the authors propose an architectural modification to the image based encoder of PQN. They apply max pooling in addition to Hadamard products to create what they deem "Hadamax." The authors then evaluate their approach on the Atari video game dataset. They compare against regular PQN as well as the Rainbow algorithm, showing improvement over baselines. Furthermore, they also perform an ablation analysis showing the ablative effects of max pooling, Hadamard operations, and and GELUs.

**Questions:**

1. Can you please elaborate on the technical novelty of the approach?

2. How could the approach be applied to different domains or model architectures? Do you believe that performance improvements may be gained with other datasets/models?

**Ethical Concerns:**

["NO or VERY MINOR ethics concerns only"]

**Final Justification:**

The authors have addressed my concerns in the rebuttal. I therefore upgrade my score to 4.

**Limitations:**

Yes, the authors have discussed limitations in Conclusions and Future Work.

**Paper Formatting Concerns:**

None.

**Quality:**

3

**Strengths And Weaknesses:**

Strengths:

1. The approach is very simple and easy to implement. I believe that I would have few issues attempting to reproduce the results in the paper.

2. The paper is well written, clear, and easy to understand.

3. Performance over baselines is quite impressive considering the simplicity of the approach (an architectural change to the image encoder).

Weaknesses:

1. The scope of the evaluation is very narrow. The experiments focus only on Atari-57; there are no other domains used for evaluation. It is thus difficult to understand how well the performance improvement could generalize across other datasets.

2. The technical novelty is quite limited. The approach is largely a tweaking of the image encoder within the existing PQN framework.

3. The scope of models is also narrow. It is unclear how this approach could improve image encoders for other RL architectures or even encoders for other tasks such as image recognition.

---

> ### Author Rebuttal · Authors · 2025-07-31
>
> Thank you for your detailed review and for highlighting the strengths: the simplicity, reproducibility, clarity, and impressive performance from architectural changes. We would like to respond to the weaknesses and questions raised.
>
> - We understand that the ALE 200M benchmark might seem like a limited evaluation. However, we would like to kindly reiterate that we have performed experiments on all of the 57 games for 200M frames in the Atari suite, similar to the original DQN, DDQN, C51, Rainbow papers etc. This still remains the largest, most diverse and challenging testbed in visual (discrete) RL. However, in order to increase environment diversity and take your concerns seriously, we have ran additional preliminary experiments on another pixel-based environment “VizDoom: Deathmatch”, using the actor-critic PPO algorithm:
>
> | VizDoom Deathmatch | 10M | 20M | 30M | 40M |
> |---------|--------------|-------------|--------------|---------------|
> | Hadamax | **-1.62** ± 0.20 | **8.08** ± 1.61 | **19.36** ± 1.82 | **29.38** ± 10.31 |
> | Baseline | -2.81 ± 0.40 | 1.27 ± 0.80 | 3.10 ± 0.89 | 5.21 ± 1.27 |
>
> Also here, we show a remarkably strong increase over the baseline by replacing the encoder. We will add the results to our updated manuscript, extending our testbed beyond Atari.
>
> - The Reviewer has a point in that there is no thorough technical analysis or proof as a precursor to the encoder design. The Hadamax design has been the product of a generalist look on all of the recent work in non-algorithmic RL, in order to strongly improve the widely used, older ‘nature’ encoder that has been used in pixel-based RL since 2015. To the best of our knowledge, the only alternative encoder used in literature was the Impala encoder. We therefore believe that we bring novelty in the way that encoders are structured, also showing that deeper encoders such as Impala are not necessarily needed for the state-of-the-art performance gains that Hadamax can bring. Adding to this, we have now done additional experiments further analyzing the effect of the isolated Hadamard and the Maxpooling operators on the convolutional feature space. Next to the effective rank already discussed in the paper, we have analyzed the feature space channel cosine similarity, as done in [1]. A low cosine similarity will indicate that convolutional channels are extracting uncorrelated features from the input data, which is positive shows a feature diversity. Maxpooling has shown to increase effective rank but also slightly increases the channel cosine similarity. Meanwhile, the addition of the Hadamard component shows a strong decrease in channel cosine similarity by up to **90**%. This gives further insights into why combining Maxpool and Hadamard works so well, as the Hadamard interaction compensates against the increase in channel cosine similarity caused by Maxpooling. We will make sure to add these results to the main paper, strengthening  our reasoning behind the proposed architecture. A table summarizing the results is seen below:
>
> | Metric                          | Baseline | + Maxpool       | + Hadamard              | Hadamax                          |
> |---------------------------------|----------|--------------------------|----------------------------------|----------------------------------|
> | Convolutional Effective Rank    | Base     | +10%              | +10%                    | **+10-20%**      |
> | Channel Cosine Similarity       | Base     | +20%      | **-90%**   | *-50%*  |
>
> - In our paper (fig. 7) we show that Hadamax is also effective on the distributional C51 architecture. Next to this, we have now shown Hadamax to also be working on the PPO actor-critic architecture on VizDoom. We therefore believe that this approach can be applied to different domains and model architectures, as long as there is convolutional downsampling involved. Besides this, we believe that the practical effectiveness of the Hadamax encoder might also allow researchers working in e.g. Image Recognition to try and implement these architectural changes in their research on encoder synthesis.
>
> Nonetheless, we thank you for the review on our work. We would however kindly ask to reconsider the given score, as we have now added strong results on an additional hard pixel-based environment (Vizdoom) in the paper, as well as more insights into the effect of Maxpooling and Hadamard on the convolutional features. Furthermore, we would like to emphasize that we believe we have made a very strong academic contribution in encoder synthesis in pixel-based RL. Most recent algorithms are still employing the ‘nature’ DQN encoder from 2015- we believe this indicates that successful encoder synthesis for Reinforcement Learning is very hard, due to the unstable nature of optimization as compared to Supervised Learning. To the best of our knowledge, Hadamax-PQN is now a state-of-the-art RL algorithm on Atari when looking at wall-clock time and performance.
>
> [1] Manifold Regularized Dynamic Network Pruning, Tang et al, 2021.

---

> > ### Comment · Reviewer_PtqA · 2025-08-07
> >
> > Thank you. The rebuttal has addressed some of my concerns.

---

### Official Review · Reviewer_J3f7 · 2025-07-03

**Clarity:** 4
**Significance:** 3
**Originality:** 3
**Rating:** 4
**Confidence:** 4

**Summary:**

This paper introduces a new network architecture for pixel-based model-free RL called Hadamax encoder. It replaces ResNet as the visual encoder in the PQN algorithm. It achieves a new SOTA performance on Atari-57, while being much faster in terms of computation.

**Questions:**

- Will such an architecture change also improve performance in e.g. standard computer vision benchmarks? Can it compete with the newest visual encoders? Why or why not?

**Ethical Concerns:**

["NO or VERY MINOR ethics concerns only"]

**Final Justification:**

The rebuttal addressed most of my concerns, I still believe the paper can further benefit from more analysis, but I also see the paper has pretty strong empirical result and is well-written. The new results in the rebuttal are also helping to strengthen the paper's significance. I maintain my score of 4: Borderline accept.

**Limitations:**

Yes

**Quality:**

3

**Strengths And Weaknesses:**

Strengths

**Quality**
- Paper written with high quality,
- It is nice the performance improvement can be achieved by only changing the network architecture.
- The insights such as that max-pooling might be overlooked in RL, "average pooling achieves the opposite" are interesting
- Ablation is nice.
- Many technical details are provided.

**Clarity**
- Paper is overall clear and easy to follow.

**Significance**
- The resulting performance and wall-clock time improvement are quite significant and convincing.

**Originality**
- While the proposed changes are relatively simple, it is novel to show that they can lead to very significant performance gain, which also brings new insights.

Weaknesses

Major:
- Paper can greatly benefit from further analysis of why certain components such as max-pooling can help so much? Currently there is only empirical result and the arguments are entirely based on the performance comparisons. Does it provide better visual features? Does it make training more stable?


Minor:
- The proposed method is faster but does it also take less computation per forward pass? I would suggest in the intro mention whether Hadamax encoder is faster per forward pass, or is it slower per forward pass but finish training at a smaller number of steps and thus leads to a smaller wall-clock time. Good to show comparison of total computation.
- Can benefit from testing on more benchmarks such as any MuJoCo benchmarks with visual input. Less important. The results on Atari are already quite convincing and might be too much work.

---

> ### Author Rebuttal · Authors · 2025-07-31
>
> We sincerely thank you for your supportive review. Your appreciation for the writing quality, insightful components (e.g., max pooling's overlooked role), thorough ablations, and performance gains is encouraging. We will respond to your points made.
>
> - We agree with your point of looking beyond performance analysis for certain components. As a result, we have done additional experiments analyzing two feature metrics for the isolated Hadamard and Maxpooling components. In addition to the effective rank, we have now also analyzed convolutional channel cosine similarity, as done in [1]. A low cosine similarity indicates that the channels are extracting dissimilar or uncorrelated features from the input data, which is desirable and suggests diversity among the channels.
> The results are interesting. Our paper already shows increased effective rank (Fig. 8) in the deeper convolutional feature space (layer 2 & 3) as compared to the baseline. However, investigating both Hadamard and Maxpooling separately shows that Maxpooling increases effective rank as compared to the baseline, but also slightly increases channel cosine similarity. Meanwhile, the Hadamard component also slightly increases effective rank but massively decreases channel cosine similarity by up to 90% as compared to the baseline. This gives further insights into why the combination of Maxpooling and Hadamard could work so well, as Hadamard strongly compensates against the increase in channel cosine similarity caused by Maxpooling. We will add these results to the main paper, along with a detailed explanation. A table summarizing the results is seen below:
>
> | Metric                          | Baseline | + Maxpool       | + Hadamard              | Hadamax                          |
> |---------------------------------|----------|--------------------------|----------------------------------|----------------------------------|
> | Convolutional Effective Rank    | Base     | +10%              | +10%                    | **+10-20%**      |
> | Channel Cosine Similarity       | Base     | +20%      | **-90%**   | *-50%*  |
>
> - We have analyzed the forward pass computation time for both Hadamax encoder, baseline encoder, and the Rainbow encoder. The inference times are as follows:
>
>
> | Architecture | Inference time (seconds) |
> |----------------|----------------|
> | Rainbow | 0.00059 |
> | PQN | 0.00039 |
> | PQN (Impala) | 0.00140 |
> | Hadamax-PQN | 0.00175 |
>
> This shows that due to setting the convolutional stride to 1, and adding maxpooling elements as well as Hadamard interactions, the inference time of the encoder will increase. Hadamax inference time is slightly higher than Impala inference time. We will be sure to add these inference times to the main paper in the camera-ready version.
>
> - We are grateful for the reviewer acknowledging the scope of the Atari-57 benchmark. We have however ran an additional evaluation on a PPO baseline in the pixel-based Vizdoom environment: Deathmatch. This encoder was similar to the Atari encoder, which allows us to directly implement the Hadamax encoder here. The results are as follows:
>
> | VizDoom Deathmatch | 10M | 20M | 30M | 40M |
> |---------|--------------|-------------|--------------|---------------|
> | Hadamax | **-1.62** ± 0.20 | **8.08** ± 1.61 | **19.36** ± 1.82 | **29.38** ± 10.31 |
> | Baseline | -2.81 ± 0.40 | 1.27 ± 0.80 | 3.10 ± 0.89 | 5.21 ± 1.27 |
>
> The results show a significant increase over the baseline encoder, and will be added to the camera-ready version to improve the environment diversity.
>
> - We have not run any tests in standard computer vision benchmarks. We do think that the Hadamax architecture, or maybe a revised, more complex version thereof, can also improve performance in standard computer vision benchmarks. However, this is always hard to predict, because if we look at the other way around, the newest computer vision encoders generally do not transfer well to pixel-based RL. This is a part of the reason that a new state-of-the-art algorithm such as PQN still uses the 'nature' DQN encoder from 2015.
>
> Thank you again for your insights. We will implement the more detailed convolutional feature analysis, the VizDoom results and the inference times in our manuscript, in order to strengthen the camera-ready version of the paper.
>
> [1] Manifold Regularized Dynamic Network Pruning, Tang et al, 2021.

---

### Official Review · Reviewer_GuDX · 2025-07-03

**Clarity:** 3
**Significance:** 4
**Originality:** 3
**Rating:** 5
**Confidence:** 4

**Summary:**

This paper introduces a new network architecture for RL on image-based inputs (Hadamax), which builds on PQN by using GELU activations, Hadamard products, and max pooling. The author(s) evaluate on Atari-57, comparing with and significantly outperforming Double DQN, C51, Rainbow, and PQN. Using Atari-10 the author(s) also show that Hadamax architecture is similarly effectively applied on top of C51, demonstrating that its benefits are not restricted specifically to PQN or the parallel training setting more generally.

Effects on the representation learned at various layers of the network are empirically analyzed and an ablation study is performed moving between PQN and Hadamax PQN.

**Questions:**

* What do you think the biggest weakness or limitation of Hadamax is?
* Could you explain why the contributions would necessarily be specific to pixel-based inputs?I suppose this is more a question about the Hadamard product component, but nonetheless I’ll rephrase as: Do you think a related approach could similarly benefit non-pixel-based domains and if not why is your reasoning?

**Ethical Concerns:**

["NO or VERY MINOR ethics concerns only"]

**Final Justification:**

Strong paper and continue to advocate for acceptance. The rebuttal slightly raised my estimation and will discuss with reviewers.

**Limitations:**

yes

**Quality:**

4

**Strengths And Weaknesses:**

Strengths:
* As this is essentially “just” a network architecture change, and would be quite simple to implement and reproduce, the magnitude of performance improvement is extremely strong. This is two strengths, (1) very simple to implement / reproducible and (2) strong empirical performance.
* The empirical work is also quite thorough, comparing with multiple baselines in the primary results and conducting well-chosen secondary experiments to supplement those results.
* The paper is well written and easy to follow.

Weaknesses:
These are really not significant weaknesses, and more just potential areas for improvement.

* This is such a generally applicable architecture that not demonstrating it on more domains is arguably the biggest weakness, and missed opportunity, in the paper. This is simply because only evaluating on Atari begs the question if this is somehow especially well suited to that one domain. At the same time, when others try it on other image-based domains and (I expect) find that it does work, that finding won’t really get published. And as a result, in the best case scenario where this is the new architecture everyone wants to use, the knowledge that it works broadly will get delayed much longer because it wasn’t communicated in a published paper. If this gets in, please include in your talk/poster additional domains to get the word out.
* After writing the above rant I realize the same logic applies to the type of base algorithm used with Hadamax: I would really like to see this used with an actor-critic / policy gradient algorithm.
* There is some degree of muddying the waters with the focus in the writing on vectorized RL. I don’t think that is the main point, nor should it be entirely downplayed, but because of how the paper is structured it leads the reader in that direction for long enough to distract from the core contributions.
* Figure 2 is great and the math is clear enough to implement, but I would have benefitted from another diagram/visual to dive deeper into the Hadamard product followed by max pooling component.

---

> ### Author Rebuttal · Authors · 2025-07-31
>
> Thank you for your positive review—we really appreciate your high rating and enthusiasm for the simplicity, reproducibility, and strong empirical performance of Hadamax. We'll address your stated weaknesses and questions below:
>
> Weaknesses:
>
> - We agree with the reviewer that implementations across different image-based domains would present more opportunities in terms of the academic contribution and transfer of knowledge. However, as much as we would like to have included, for example, the full pixel-based MuJoCo domain, the compute requirements are a limiting factor. The full 57 game Atari suite for 200M frames (as is used in the original DQN, DDQN, Rainbow, C51 papers) is already a very demanding benchmark.
>
> Having said this, we do see that environment diversity is a recurring theme amongst the reviewers and wanted to address some of these concerns. We have therefore conducted preliminary experiments using the **actor-critic** based PPO, on a pixel-based VizDoom (Deathmatch) environment for 40M frames:
>
>
>
> | VizDoom Deathmatch | 10M | 20M | 30M | 40M |
> |---------|--------------|-------------|--------------|---------------|
> | Hadamax | **-1.62** ± 0.20 | **8.08** ± 1.61 | **19.36** ± 1.82 | **29.38** ± 10.31 |
> | Baseline | -2.81 ± 0.40 | 1.27 ± 0.80 | 3.10 ± 0.89 | 5.21 ± 1.27 |
>
> Note that, as in the PQN framework, we only substituted the original encoder with a Hadamax encoder. Although Vizdoom is usually trained for longer, there is already a clear performance boost over the baseline encoder, and we will add the results, as well as the code, in the camera-ready version.
>
> - As for your comments on the poster/presentation, if our work gets accepted, we will be sure to try and include as many use-cases as possible.
>
> - About the emphasis on vectorized RL: We see your point and apologize if this muddied the waters. We'll slightly revise the structure of the manuscript for a clearer storyline in the camera-ready version.
>
> - We have already included the precise code of the encoder in Appendix B. Or do you mean that readers could have benefited from it in a more visual way? If it’s the latter, we are more than willing to create a more in-depth diagram for the camera-ready version.
>
> Questions:
> - The biggest limitation of Hadamax is that it increases the parameter size of the encoder and adds additional training time, in comparison to a less complex baseline encoder. Furthermore, the Hadamax encoder, in this form, has been designed for pixel-based environments only where a CNN feature extractor is followed by a bottleneck (Linear layers).
> - We think that, because pixel-based inputs (especially in Atari) are high-dimensional, the encoder synthesis differs from that of non-pixel domains. We assume Hadamax works so well here because it improves the feature extraction process. In non-pixel-based domains, the hidden layers are commonly larger than the original input spaces, so in that case we see the MLP encoder as more of an ‘optimization vessel’ rather than a true (compressing) encoder. From what we’ve seen, these MLP encoders on state-based inputs generally have different design requirements.
>
> We thank you once again for your detailed review, and praise of our work. We will implement your suggestions as well as the VizDoom results in the main paper, in order to strengthen the camera-ready version.

---

### Decision · Program_Chairs · 2025-09-17

**Decision:**

Accept (poster)

**Comment:**

The reviewers agree that the paper presents surprising empirical evidence that small tweaks to the existing PQN algorithm can result in significant performance increases. They found the paper to be clearly presented and the hypotheses to be well-evaluated.